# Impact of Salvage Surgery on Health-Related Quality of Life in Oral Squamous Cell Carcinoma: A Prospective Multi-Center Study

**DOI:** 10.3390/jcm12206602

**Published:** 2023-10-18

**Authors:** Sven Zittel, Julius Moratin, Sinclair Awounvo, Thomas Rückschloß, Kolja Freier, Oliver Ristow, Michael Engel, Jürgen Hoffmann, Christian Freudlsperger, Dominik Horn

**Affiliations:** 1Department of Oral and Cranio-Maxillofacial Surgery, University of Heidelberg, Im Neuenheimer Feld 400, D-69120 Heidelberg, Germany; julius.moratin@med.uni-heidelberg.de (J.M.); thomas.rueckschloss@med.uni-heidelberg.de (T.R.); oliver.ristow@med.uni-heidelberg.de (O.R.); michael.engel@med.uni-heidelberg.de (M.E.); juergen.hoffmann@med.uni-heidelberg.de (J.H.); christian.freudlsperger@med.uni-heidelberg.de (C.F.); 2Institute of Medical Biometry, University of Heidelberg, Im Neuenheimer Feld 130.3, D-69120 Heidelberg, Germany; awounvo@imbi.uni-heidelberg.de; 3Department of Oral and Cranio-Maxillofacial Surgery, Saarland University Hospital, Kirrberger Straße, D-66424 Homburg, Germany; kolja.freier@uks.eu (K.F.); dominik.horn@uks.eu (D.H.)

**Keywords:** oral squamous cell carcinoma, recurrent oral squamous cell carcinoma, salvage surgery, quality of life, EORTC

## Abstract

Background: Patients with recurrent oral squamous cell carcinoma (OSCC) have limited treatment options. Salvage surgery offers potential curative therapy. The need for extensive ablative surgery together with microvascular reconstruction implies invasive and painful treatment with questionable functional outcome. To address the impact of salvage surgery on the health-related quality of life (HRQoL) of patients suffering from recurrent OSCC, a multi-center prospective analysis was initiated. Material and Methods: Patients with recurrent OSCC from 2015 to 2022 at two German cancer centers were included. Interdisciplinary tumor board decisions determined surgery as the only curative treatment modality. HRQoL, was assessed via a EORTC questionnaire (European Organization for Research and Treatment of Cancer—EORTC: QLQ-C30 and QLQ-H&N35) in dependence of the recurrent tumor stage. Patients completed the questionnaires once before surgery (baseline) and then every 3 months during follow-up or up to the end of treatment. Results: In total, 55 patients were included. The mean follow-up period was 26.7 ± 19.3 months. Global health status showed superior mean scores after 12 months (60.83 ± 22.58) compared to baseline (53.33 ± 26.41) in stage 1 and 2 recurrent tumors. In advanced recurrent tumors’ mean scores for global health showed only minor positive differences after 12 months (55.13 ± 22.7) compared to baseline (53.2 ± 25.58). In terms of the mouth pain, mean scores were lower after salvage surgery in small recurrent tumors after 12 months (20.37 ± 17.73) compared to baseline (41.67 ± 33.07; Wilcoxon two-sample signed-rank test *p* = 0.028). In advanced recurrent tumors, a significant reduction in mean scores was detected 3 months after salvage surgery (29.7 ± 22.94) compared to baseline (47.76 ± 25.77; Wilcoxon two-sample signed-rank test *p* = 0.003). Up to 12 months, swallowing function was evaluated inferior compared to baseline independent of tumor stage (Mean score recurrent stage I/II: 12-months 48.15 ± 27.57, baseline 28.7 ± 22.87; stage III/IV: 12-months 49.36.42 ± 27.53; baseline 30.13 ± 26.25). Conclusion: Improved HRQoL could be obtained in advanced recurrent OSCC after salvage surgery despite reduced swallowing function. In small recurrent tumors, overall, HRQoL was superior to baseline. Salvage surgery positively affected pain burden. For advanced recurrent tumors, important pain relieve could be observed as soon as 3 months after surgery.

## 1. Introduction

The recurrence of OSCC occurs in up to 50% of patients with advanced primary tumors independent of the initial oncologic therapy [1,2,3,4,5]. Health-related quality of life (HRQoL) is reciprocally dependent on tumor progression [6,7]. Primary surgery followed by radiochemotherapy is considered as the standard of care in advanced OSCCs [8]. In case of recurrence, treatment options are limited. Re-radiation approaches have dose limits due to tissue toxicity, especially in early cancer recurrence. In pretreated recurrent OSCC, the radiation dose has usually been exhausted in the primary therapeutic approach. Therefore, surgical therapy (salvage surgery) remains the treatment of choice if curation is intended [9,10,11,12]. In inoperable patients, re-irradiation can be performed for local control [11,13,14]. Recently, palliative immunotherapy has shown superiority over platin-based regimens in terms of oncological aspects, as well as the outcome of HRQoL [15,16,17]. Nevertheless, response rates to immunotherapy are low and unpredictable [18,19]. There are limited data of oncological outcome together with HRQoL in salvage surgery in OSCC after multimodal treatment failure [20]. Global HRQoL has a huge impact on patients suffering from unfavorable oncological courses [21]. The perception of pain is directly linked to the patient’s QoL [22,23,24]. In clinical practice, patients and oncologists are confronted with the tough, individual decision of whether potentially curative, but extensive, oncological surgery together with microvascular reconstruction or palliative systemic immunotherapy is appropriate [5,20,25,26]. Patients and medical experts usually link cancer surgery to painful treatment approaches resulting in poor function. Great fears are related to pain, swallowing functioning and loss of speech. The surgical assessment on these aspects is usually based on personal expertise or retrospective cohort analyses mainly focusing on technical outcomes rather than patient-specific functional and emotional outcomes. Considering the resected tissue mass in the functional and esthetic sensitive orofacial region, these conclusions are comprehensible in subjective evaluation [27]. However, the already painful destructive expansion of the tumor, resulting in progressive loss of function, is often underweighted in objective evaluation [28]. Therefore, patients expect more pain and loss of function than is experienced than vice versa. Furthermore, modern microvascular reconstruction has undergone evolution over decades, resulting in variability in microvascular transplants and virtually planned and CAD-CAM-manufactured surgical supply; therefore, fastened surgery and safe anesthesiological support are required [20,25]. In orchestration with intensive care, oncological nursing care and functional therapies, standardized treatments and rehabilitation workflows have been implemented in oncological centers in recent decades. Nevertheless, high-level evidence in salvage surgery is difficult to establish, which is why HRQoL data do not exist or are only available in small cohorts [20]. In order to strengthen the evidence, we enrolled patients in our prospective multicenter observational study. The purpose was to investigate the impact of salvage surgery on HRQoL in accordance with tumor stage, where individual, functional, technical and emotional aspects merge.

## 2. Materials and Methods

### 2.1. Study Design

A prospective multi-center study, which was confirmed by the local ethical committees of the participating centers (Heidelberg: S-186/2015; Homburg: 262/19), was conducted. Study registration was implemented (German Clinical Trials Register (DRKS) (Reg.-Nr.: DRKS00009255)). Patients were enrolled by the University Department of Maxillofacial Surgery in Heidelberg and the University Department of Maxillofacial Surgery at the Saarland Medical Centre. In total, 55 patients received salvage surgery between 2015 and 2022 (Table 1).

### 2.2. Inclusion/Exclusion Criteria

All patients had undergone multiple previous therapeutic approaches, including surgery and or radio-/chemotherapy, due to oral cancer and, thus, were defined as “heavily pre-treated”. Followed by a guideline-based staging-examination, salvage surgery was determined as curative therapy via an interdisciplinary oncological case discussion in the corresponding center. Alternative curative radiochemotherapy was not considered in the tumor board’s decision. Exclusion criteria were the infiltration of the common and/or internal carotid artery, skull base tumors, distant metastasis and an age younger than 18 years.

### 2.3. Quality of Life

The HRQoL was evaluated using a quality of life (QoL) questionnaire of the EORTC (European Organization for Research and Treatment of Cancer—EORTC: QLQ-C30 and QLQ-H&N35). This is a widely used questionnaire consisting of a basic questionnaire and an associated questionnaire specially adapted for head and neck cancers [29,30]. Patients completed the questionnaires once before surgery (baseline) and then every 3 months during follow-up or at to the end of treatment. High score values for global health status (GHS) indicate a good QoL. In contrast, high score values for symptom-scores, such as pain in the mouth and swallowing, indicate high-level symptoms. CT scans of the head/neck and thorax were performed every three months together with clinical follow-up every 4 weeks.

### 2.4. Statistical Analyses

Statistical analysis was performed using R version 4.2.2. Clinical and histo-pathological parameters were assessed descriptively. Metric variables were described using mean and standard deviations. Overall survival and disease-specific survival were assessed using Kaplan–Meier estimation. Comparison of overall survival between the small and advanced recurrent tumors groups was performed using the Log-rank test. The QoL dataset was evaluated according to the scoring manual (Third edition from 2001) [29,31]. Mann–Whitney’s U test (MWU) was used to compare the average scores between the small and advanced recurrent tumors groups at a given time point. For a given group, average score comparison between a given time-point and baseline was conducted using the Wilcoxon two-sample signed-rank test. All statistical tests were conducted to a 5% significance level, and *p*-values were reported. Additionally, a 95% confidence interval (CI) for the Hodges–Lehmann estimator was reported for the MWU test. Overall group comparison was performed by means of a linear mixed-effects regression. The models included the change score as the dependent variable and the group, gender and age as independent variables. For the estimated effects, *p*-values and 95% confidence intervals were reported. Score value differences of 5 to 10 points were considered a small change, 11 to 20 points a moderate change and more than 20 points a major change [32]. Score-value differences of 10 score points or more were considered a clinically relevant change [33]. All p-values had a purely confirmatory value and were considered descriptively. An adjustment based on multiple testing was not performed. P values of less than 0.05 were considered significant. Missing values were not imputed.

## 3. Results

### 3.1. Basic Patient Characteristics

In total, 55 intensively pretreated patients with recurrent cancer were enrolled and received salvage surgery in the participating centers from 2015 to 2022. Moreover, 49 patients were treated at the Department of Maxillofacial Surgery in Heidelberg, and 6 patients were treated at the Department of Maxillofacial Surgery at the Saarland Medical Centre. The mean follow-up was 26.7 months SD ± 19.34. The cohort consisted of 24 (43.6%) women and 31 (56.4%) men. The mean age was 65.89 SD ± 10.41 years. The response rate for the QoL Questionnaires was 77.4% after 12 months (Table 1).

### 3.2. Clinical Description of the Cohort

All patients received mean 3.2 SD ± 1.25 prior to treatment modalities. In total, 28 (50.9%) patients suffered from a secondary cancer, 23 (41.8%) patients showed a local recurrence and 4 (7.3%) patients presented with regional lymph node metastasis but stable local tumor control. Except for 12 (21.8%) patients, all resected specimens showed clear margins in histopathological examination. Moreover, 50 (90.9%) patients received microvascular reconstruction, while 5 (9.1%) patients received local flap or pedicled flap reconstruction. Two (3.6%) flaps were lost due to venous stasis. In total, 31 (56.4%) patients required a permanent tracheostomy, and 23 (41.8%) patients received a gastric percutaneous endoscopic gastrostomy tube. Of these, 5 (21.7%) patients were at stages 1 and 2, and 18 (78.3%) patients were at stages 3 and 4 (Table 1).
jcm-12-06602-t001_Table 1Table 1Clinical data and histopathological parameters of the cohort.Variable*n*Patients
Heidelberg49Homburg06Return of Questionnaires after 12 months77.4%Age65.89 SD ± 10.41Sex
female24 (43.6%)male31 (56.4%)Primary Treatment
Surgery13 (23.6%)RCT ^1^5 (9.1%)Surgery + RCT33 (60.0%)Surgery + RT ^2^4 (7.3%)Pre-salvage received modalities ^3^3.2 SD ± 1.25Tumor Type
Local recurrence23 (41.8%)Secondary Tumor ^4^28 (50.9%)Regional Progress 04 (7.3%)Recurrent Stage ^5^
I10 (18.2%)II07 (12.7%)III06 (10.9%)IV32 (58.2%)Tumor Localization
Upper Jaw and palate4 (7.3%)Lower Jaw, Floor of the mouth and tongue34 (61.8%)Neck17 (30.9%)R-status
R043 (78.2%)R112 (21.8%)Reconstruction
Local wound closure05 (9.1%)Free flap50 (90.9%)Free flap failure 
Yes02 (3.6%)No53 (96.4%)Tracheostomy31 (56.4%)Gastric Percutaneous Endoscopic Gastrostomy Tube23 (41.8%)Stage I and II5 (21.7%)Stage III and IV18 (78.3%)^1^ RCT = radio-chemotherapy; ^2^ RT = radiotherapy; ^3^ Only oncological treatments with curative intent were counted as therapy; ^4^ tumors that occurred for more than 60 months after the treatment of the primary tumor or had a clearly definable localization were classified as secondary carcinomas; ^5^ recurrent tumors were categorized according to UICC classification [9,34].

### 3.3. Quality of Life

QoL data were analyzed according to the EORTC protocol.

#### 3.3.1. Global Health Status (GHS)

Small recurrent tumors (Stage 1–2) showed higher mean score values (49.40 ± 28.95; *n* = 14) at baseline, in contrast to advanced tumors (Stage 3–4) (47.31 ± 24.00 *n* = 31). In early recurrent tumors mean score values decreased three months after salvage surgery (46.15 ± 22.98; *n* = 13). Up to 9 months mean values were stable (Table 2, Figure 1). At 12 months, higher but not meaningful score values compared to baseline were detected (mean score difference to baseline (MDB) 7.5 ± 35.24; *n* = 10). In contrast, patients with advanced recurrent tumors showed an immediate increase in their mean scores three months after surgery (53.53 ± 22.13; *n* = 26). Over the following 6 months, mean scores showed a minor decrease (Table 2, Figure 1). Mean scores were superior to baseline one year after salvage surgery (MDB 1.925 ± 33.01; *n* = 13). Observed changes in mean scores in GHS were not significant (Table 2). The results of the linear mixed-effects regression analysis show that for two patients of the same age and sex with different tumor stage, the patient with the advanced recurrent tumor has a 12.4-point higher change score compared to the patient with the small recurrent tumor. Further, women have 6.5-point lower change scores compared to men and a one-year increase in a patient’s age corresponds to a one-point increase in their change score (Appendix A).

#### 3.3.2. Pain in the Mouth (H&N35)

Next, the item pain perception was further examined as a relevant clinical parameter. Baseline score values in stages 1–2 and 3–4 were on an almost equal level (stage 1–2: 48.81 ± 30.11; stage 3–4: 47.31 ± 25.59). In small recurrences, a constant decrease from baseline (48.81 ± 30.11) for up to 12 months (21.97 ± 17.19) was detected (Figure 2, Table 3). The mean difference indicates a significant pain reduction over 12 months (MDB −21.3 ± 30.36; *p* = 0.028) (Table 3). In stage 3–4 tumors, a significant decrease in mean scores immediately after salvage surgery was observed (MDB −18.75 ± 28.15; *p* = 0.003). The mean score was further reduced after 6 months (MDB −19.44 ± 29.15; *p* = 0.013). At 9 months, the score decrease was not so pronounced as in the months before but was still meaningful compared to baseline (MDB −8.854 ± 40.54; *p* = 0.379). After 12 months, we observed a clinically relevant reduction in pain scores compared to baseline (MDB −14.10 ± 41.30; *p* = 0.3197). The stage-specific means scores at each time point differed in terms of relevance but were not statistically significant (Table 3). Linear mixed-effects regression analysis shows that the change score for patients with advanced recurrent tumors is 13.5 score points lower compared to a patient with small recurrent tumors when both are of the same sex and age. Additionally, women have a 6.9-point higher change score than men, and a one-year increase in patient’s age corresponds to a 0.9-point decrease in change score (Appendix A).

#### 3.3.3. Swallowing (H&N 35)

In the analysis of the subscore swallowing, stage 1–2 tumors showed non-significant lower score values (34.52 ± 22.85; *n* = 14) in contrast to stage 3–4 tumors (37.10 ± 27.46; *n* = 31). Three-month postoperative mean scores increased in both groups (MDB stage 1–2: 28.47 ± 26.70; stage 3–4: 17.75 ± 32.60) (Table 4, Figure 3). Three-month postoperative scores were higher in stage 1–2 tumors (58.97 ± 28.35; *n* = 13) in contrast to advanced tumors (53.31 ± 35.28; *n* = 26). After 12 months, mean scores values were still higher than baseline in both groups (MDB stage 1–2: 19.45 ± 23.94; stage 3–4: 17.42 ± 26.21) (Table 4, Figure 3). The mean score difference from each timepoint to baseline is shown in Table 4. The linear mixed-effects regression analysis shows that for two patients of the same sex and age with different tumor stages, the patient with the advanced recurrent tumors has a 1.1-point lower change score than the patient with the small recurrent tumors. In addition, women have a 8.5-point lower change score than men, and a year increase in the patient’s age corresponds to a 0.3-point decrease in change score (Appendix A).

#### 3.3.4. Oncologic Outcome

Recurrence was observed in 36 (65.5%) patients. In total, 25 (45.5%) patients died during follow-up. The mean follow-up time was 26.7 ± 19.3 months. The median overall survival was 33.9 months (Appendix A). The median time to re-recurrence was 11.1 months CI (5.1:16.1). Patients with small recurrent tumors (Stage 1–2) showed superior overall survival compared to those with advanced recurrent tumors, although not to a significant extent (Log rank *p* = 0.054) (Figure 4). Median overall survival (OS) was 46.4 CI (12.2:-) months for stage 1 and 2 recurrences and 20.3 CI [10.3:-] for stage 3 and 4 recurrences (Figure 4). During follow-up, 36 (65.5%) patients received additional oncological therapy in the form of further surgery, additive re-radiation, chemotherapy, immunotherapy or appropriate supportive care.

## 4. Discussion

The treatment of recurrent OSCC remains a major challenge [20,26]. Both patients and clinicians are in a complex situation, where decisions with far-reaching consequences have to be made [5,35]. Therefore, surgeons have to evaluate the technical possibility of extensive surgery against objective oncological outcome, as well as patient-specific QoL.

### 4.1. Health-Related Quality of Life (HRQoL)

The HRQoL was assessed using the European Organization for Research and Treatment of Cancer (EORTC), a widely used quality-of-life questionnaire consisting of a basic questionnaire and an associated questionnaire specially adapted for head and neck diseases [29,30]. In recurrent OSCC, the tumor stage has a huge impact on surgical decisions. Tumor size, as well as the location, determines the extent of resections and ultimately influences function. Therefore, our analysis focused on the stage-dependent impact on HRQoL during surgical salvage therapy and the corresponding follow-up. The EORTC QLQ C30 core questionnaire and H&N 35 module cover various subscores. In order to maintain clarity and comparability, only the most relevant scores, namely “Global health status” (GHS), “Pain in the mouth” (Pain) and “Swallowing”, were highlighted. We followed the evaluation methods of previous oncological studies [16]. The survey of the HRQoL in patients with advanced tumor diseases is often associated with the limited returning of questionnaires. Even in large trials, respond rates of only up to 44% were reported [36]. We were pleased to have a response rate of 77% after 12 months of follow up, which was probably due to the small study population and, therefore, the possibility for a closer patient contact. Due to the small study population, a limited number of questionnaires were available for evaluation. We observed a low response rate to questionnaires in patients who received further treatment in another discipline due to further recurrences.

In terms of GHS, small recurrent OSCC patients had stable mean scores over time (Table 2). A comprehensible decrease three months after surgery recovered over the observational period and ended 10 points above baseline. Patients suffering from advanced tumors did not show mean a score decrease immediately after surgery. The means scores slightly varied over months and ended above the baseline at 12 months. The loss of HRQoL during tumor progression can be explained by the loss of function, pain and stigmatization. Nevertheless, the differences between both groups were not significant. The mean differences in both groups from baseline to each postoperative timepoint were also proven not to be significant. These GHS findings imply that salvage surgery is a reasonable approach for locoregional recurrence when it comes to quality of life independent of recurrent tumor stage. Besides the negative impact of salvage surgery on QoL, it must be emphasized that in the context of drug-based tumor therapy, e.g., in the Keynote 048 study, comparable values were observed after one year when comparing different treatment regimens [17].

Pain relief has a great influence on the HRQoL, especially in advanced tumors [37]. In both groups, pain perception was reduced over time. In small tumors, mean scores constantly decreased for 12 months. After one year, a significant pain reduction from baseline after salvage surgery could be observed (Table 3). In advanced recurrent OSCCs, significantly lower pain levels could already be observed after three months. We, therefore, see evidence of a pain-relieving benefit of surgery when it comes to recurrent OSCC. This is unlike the conventional and intuitive assumptions of laymen, as well as medical experts, for whom extensive orofacial surgery is accompanied by an expectation of progressive pain.

When it comes to swallowing function, there were no significant differences between both groups at any given timepoint. Nevertheless, swallowing function is significantly impaired after salvage surgery compared to baseline in both groups. This observation seems to be constant, as mean scores for swallowing then show only minor changes up to one year. Worth mentioning is the fact that head and neck cancer therapy in general impacts swallowing function. In a study including 109 patients with primary head and neck cancer, treated either with surgery, surgery in combination with radiotherapy or chemoradiation were assessed for their swallowing function. It was found that 75.6% of the patients were affected in their social life after oncological therapy [38]. Therefore, transparent patient education, early decisions in terms of temporary feeding support (gastric feeding tubes) and the early implementation of functional training are mandatory to ensure the ideal support of OSCC patients [39].

### 4.2. Oncologic Outcome

In a previous evaluation, salvage surgery in intensively pretreated patients showed a 1-year and 2-year disease specific survival (DSS) of 68.4% and 59.3%, respectively [20]. In a meta-analysis of 1080 head and neck cancer patients, a 5-year DDS of 39% was reported [9]. The survival rates in our cohort are comparable to the ones reported in the literature. The field of re-irradiation shows ongoing scientific research [40]. Carbon ion radiotherapy has shown promising success in selected treatment centers [11]. But based on the ratio of median survival of 33.9 months and the rapidly achieved baseline score values in GHS, extensive surgery is justified.

Salvage surgery has positive oncological influence on OSCC patients. It is a safe, technical, feasible and proven therapy option in complex oncological cases. The impact of salvage surgery is associated with future advances in the prediction accuracy of immunotherapeutic response rates. The combination of salvage surgery with immunotherapeutic approaches offers promising opportunities.

### 4.3. Limitations

Due to the single-arm study design, the low number of cases and the small number of returned questionnaires, the power of this study is limited. In addition, the subdivision into subgroups led to the problem of unbalanced groups, which further reduced the power of the performed statistical tests. Static power was insufficient for a comparison of quality of life with different primary therapy approaches. The perioperative medical history in terms of pain killers has not been considered in this study. Moreover, it has to be considered that patients with a better QoL tend to be more likely to complete questionnaires.

## 5. Conclusions

HRQoL can be maintained despite extensive surgery. In particular, salvage surgery seems to have a positive impact on the course of pain. As swallowing function seems impaired after surgery, reconstructive techniques should intensively focus on swallowing anatomy, and functional therapies should be intensively integrated into early rehabilitation.

## Figures and Tables

**Figure 1 jcm-12-06602-f001:**
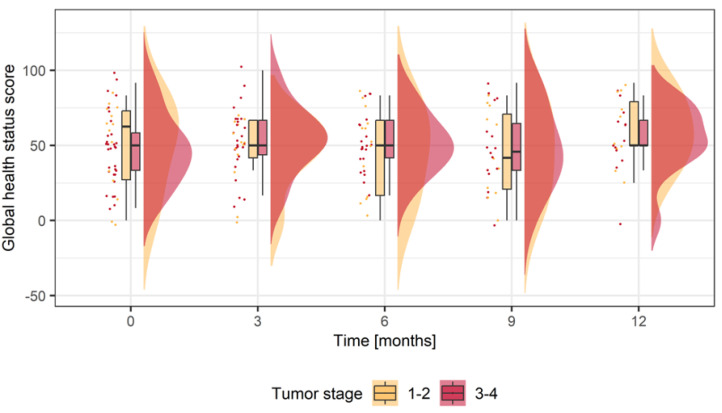
Score values for global health status in small recurrent tumors (Stage 1–2; yellow) and advanced recurrent tumors (Stage 3–4; red). Boxplots display mean scores prior to surgery (0 = baseline) and different timepoints during follow-up for up to one year (12 = months postoperatively). The dots display the factual score values (Stage 1–2; yellow; Stage 3–4; red). The curve represents the density function (Stage 1–2; yellow; Stage 3–4; red).

**Figure 2 jcm-12-06602-f002:**
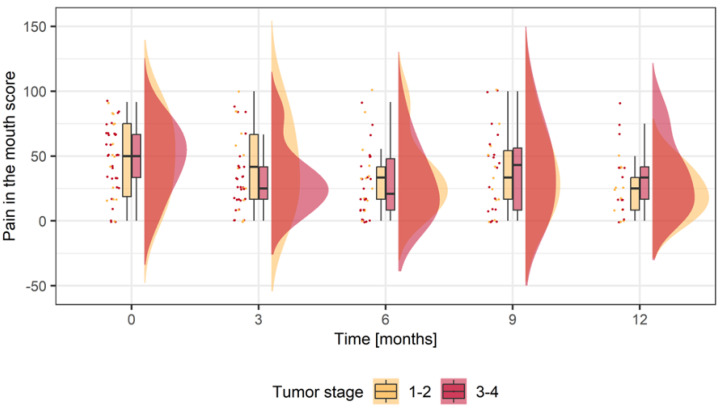
Score values for pain in the mouth in small recurrent tumors (Stage 1–2; yellow) and advanced recurrent tumors (Stage 3–4; red). Boxplots display mean scores prior to surgery (0 = baseline) and different timepoints during follow up for to one year (12 = months postoperatively). The dots display the factual score values (Stage 1–2; yellow; Stage 3–4; red). The curve represents the density function (Stage 1–2; yellow; Stage 3–4; red).

**Figure 3 jcm-12-06602-f003:**
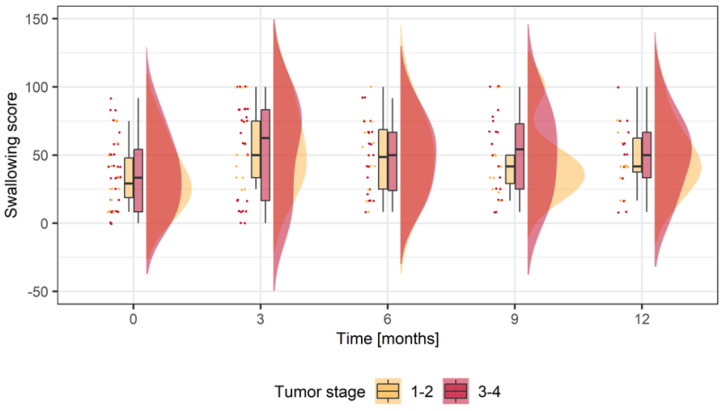
Score values for swallowing in small recurrent tumors (Stage 1–2; yellow) and advanced recurrent tumors (Stage 3–4; red). Boxplots display mean scores prior to surgery (0 = baseline) and different timepoints during follow-up for to one year (12 = months postoperatively). The dots display the factual score values (Stage 1–2; yellow; Stage 3–4; red). The curve represents the density function (Stage 1–2; yellow; Stage 3–4; red).

**Figure 4 jcm-12-06602-f004:**
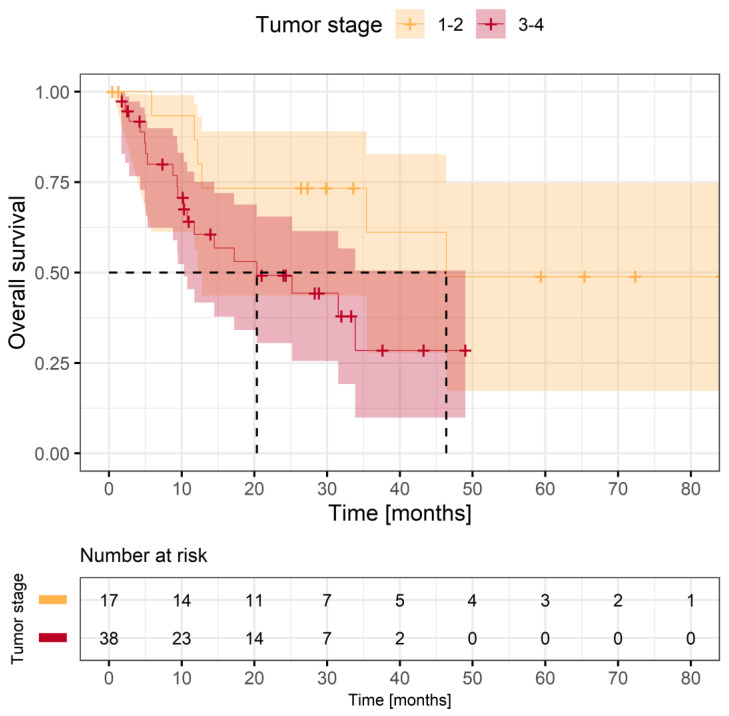
Overall survival with regard to tumor stage (Stage 1–2; yellow vs. Stage 3–4; red); Log-rank test, *p* = 0.054. Dashed line indicates median survival.

**Table 2 jcm-12-06602-t002:** Score-values for global health status (GHS) in both groups in comparison. ^1^ Mean score at each timepoint. ^2^ Mean score differences to Baseline (MDB). ^3^ MWU = Mann–Whitney’s U Test; HL = confidence interval (CI) for the Hodges–Lehmann estimator. ^4^ Wilcoxon two-sample signed-rank test.

GHS	Stage 1–2 ^1^ Mean/SD	Stage 3–4 ^1^ Mean/SD	*p* Value [CI] ^3^	Stage 1–2 ^2^ Mean/SD	*p* Value ^4^	Stage 3–4 ^2^ Mean/SD	*p* Value ^4^
Baseline	49.40 ±28.95 (*n* = 14)	47.31 ±24.00 (*n* = 31)	0.594 [−17, 25]	−	−	−	−
03 months	46.15 ±22.98 (*n* = 13)	53.53 ±22.13 (*n* = 27)	0.485 [−17, 8.3]	−10.42 ±31.21	0.327	5.335 ±24.28	0.187
06 months	46.15 ±28.59 (*n* = 14)	51.59 ±18.56 (*n* = 22)	0.760 [−25, 17]	−27.78 ±39.78	0.765	2.918 ±22.34	0.609
09 months	45.46 ±28.95 (*n* = 11)	48.44 ±26.91 (*n* = 17)	0.710 [−25, 25]	−2.272 ±35.96	0.906	−6.111 ±36.11	0.783
12 months	59.85 ±21.67 (*n* = 11)	55.13 ±22.70 (*n* = 13)	0.678 [−17, 25]	7.50 ±35.24	0.600	1.925 ±33.01	0.528

**Table 3 jcm-12-06602-t003:** Score-values for pain in the mouth in both groups in comparison. ^1^ Mean score at each timepoint. ^2^ Mean score differences to Baseline (MDB). ^3^ MWU = Mann–Whitney’s U Test; HL = confidence interval (CI) for the Hodges–Lehmann estimator. ^4^ Wilcoxon two-sample signed-rank test.

Pain	Stage 1–2 ^1^ Mean/SD	Stage 3–4 ^1^ Mean/SD	*p* Value [CI] ^3^	Stage 1–2 ^2^ Mean/SD	*p* Value ^4^	Stage 3–4 ^2^ Mean/SD	*p* Value ^4^
Baseline	48.81 ±30.11 (*n* = 14)	47.31 ±25.59 (*n* = 31)	0.834 [−17, 17]	−	−	−	−
03 months	42.95 ±33.65 (*n* = 13)	31.69 ±24.75 (*n* = 27)	0.375 [−8.3, 33]	−0.70 ±26.93	0.721	−18.75 ±28.15	0.003
06 months	35.04 ±29.52 (*n* = 13)	28.41 ±26.68 (*n* = 22)	0.470 [−17, 25]	−11.37 ±27.95	0.119	−19.44 ±29.15	0.013
09 months	37.88 ±31.26 (*n* = 11)	39.04 ±32.86 (*n* = 18)	0.946 [−25, 25]	−8.334 ±21.87	0.203	−8.854 ±40.54	0.379
12 months	21.97 ±17.19 (*n* = 11)	34.62 ±29.04 (*n* = 13)	0.380 [−33, 8.3]	−21.30 ±30.36	0.028	−14.10 ±41.30	0.197

**Table 4 jcm-12-06602-t004:** Score-values for swallowing in both groups in comparison. ^1^ Mean score at each timepoint. ^2^ Mean score differences to Baseline (MDB). ^3^ MWU = Mann–Whitney’s U Test; HL = confidence interval (CI) for the Hodges–Lehmann estimator. ^4^ Wilcoxon two-sample signed-rank test.

Swallowing	Stage 1–2 ^1^ Mean/SD	Stage 3–4 ^1^ Mean/SD	*p* Value [CI] ^3^	Stage 1–2 ^2^ Mean/SD	*p* Value ^4^	Stage 3–4 ^2^ Mean/SD	*p* Value ^4^
Baseline	34.52 ±22.85 (*n* = 14)	37.10 ±27.46 (*n* = 31)	0.824 [−17, 17]	−	−	−	−
03 months	58.97 ±28.35 (*n* = 13)	53.31 ±35.28 (*n* = 27)	0.708 [−25, 25]	28.47 ±26.70	0.012	17.75 ±32.60	0.007
06 months	49.08 ±27.56 (*n* = 13)	48.54 ±25.77 (*n* = 22)	0.953 [−19, 25]	19.17 ±22.24	0.018	16.90 ±23.42	0.004
09 months	46.72 ±28.43 (*n* = 11)	51.08 ±30.37 (*n* = 18)	0.718 [−33, 17]	17.71 ±9.385	0.058	17.81 ±28.04	0.021
12 months	48.49 ±24.95 (*n* = 11)	49.36 ±27.53 (*n* = 13)	0.815 [−25, 25]	19.45 ±23.94	0.042	17.42 ±26.21	0.052

## Data Availability

The data presented in this study are available in justified cases on request from the corresponding author.

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
