# Peer review of "Impact of Salvage Surgery on Health-Related Quality of Life in Oral Squamous Cell Carcinoma: A Prospective Multi-Center Study"

_jcm, 2023, doi:10.3390/jcm12206602_

Round 1

Reviewer 1 Report

This is a prospective case series primarily looking at quality-of-life (QOL) data in a pool of recurrent OSCC patients. The existing literature regarding this issue is limited, so the paper offers interesting findings. However, I have several concerns with the document.

1. In this patient cohort, cases with a secondary tumor accounted for 50.9%. In my opinion, tumor recurrence is not the same as a secondary tumor. So the title "health-related quality of life in re-current oral squamous cell carcinoma" is not appropriated.

2. In the introduction section, the author should simply describe the result of previous studies regarding the effect of salvage surgery on the QOL of recurrent OSCC patients.

3. The 1st paragraph in the discussion section should be integrated into the section of introduction.    

4. In the introduction section (p.2, the first line of 1st paragraph), the authors should cite relevant references to support the opinion ("Recurrence of OSCC occurs in up to 50% of patients ").

5. For the scale of EORTC: QLQ-C30 and QLQ-H&N35, the differences that are deemed clinically meaningful are defined as 10 score points or more. However, when the authors described the outcomes, for example, the Global health status, a change of 7.5 score point over time was considered as superior to the baseline. I think this finding was not clinically meaningful. The authors should emphasize the findings either clinically meaningful or statistically significant.

6. The EORTC: QLQ-C30 and QLQ-H&N35 included 65 items, why the authors only demonstrated the outcomes about the Global health status, the mouth pain and the swallowing? Did the description of the other outcomes change the conclusion of the manuscript?

Reviewer 2 Report

Nowadays, the gold standard primary treatment of recurrent oral cavity cancer is surgery, that according to the extension of the disease can be invasive and therefore impact the patient’s function and quality of life. Although survival is the primary goal of treatment in such patients, a respectable quality of life must be also an important purpose of the oncologist and ENT surgeon. In this setting the authors performed a multicentric, prospective study on patients undergoing surgery for recurrent oral cancer analyzing quality of life, pain and swallowing before and after surgery. The study is well conducted, the methodology is clearly explained, and the results are described in a detailed way. Although the study includes a small cohort of patients with therefore low statistical power, the results are interesting. In particular, the fact that both global health and pain questionnaires showed an improvement in quality of life and pain after surgery compared to before surgery, despite the invasiveness of the surgical techniques. Although the article is generally well written, the English language should be revised to improve the intelligibility of the manuscript. Furthermore, I hereby attach some specific comments divided according to the article section:

Introduction:

-       -   When the abbreviation “HRQoL” is first used in the introduction, it should be written in full length for clarity.

-        -  From “In clinical routine….” To “..vice versa”: is this supported by the literature? In such case, please cite the studies in support. Otherwise, the strength of the sentences should be softened.

-      -    The second half of the introduction is not very clear due to poor syntax. Please revise the English language and reformulate the sentences to make it clearer.

Materials and methods:

-        -  “scull-base” is a typo, please correct it.

Results:

-       -   Table 1: the types of primary treatments should be described in Table 1, e.g. surgery n,%; surgery+RT n,%; surgery+CRT n,%; RT n,%; CRT n,%.

-    -   It would be interesting to compare the outcomes (global health, pain, and swallowing questionnaires) according to the type of primary treatment (e.g., surgery vs surgery+RT vs surgery+CRT vs RT vs CRT). If not possible, this should be highlighted in the discussion section as a limitation of the study.

Although the article is generally well written, the English language and in particular the syntax should be revised to improve the intelligibility of the manuscript. 

Reviewer 3 Report

Dear Authors,

The manuscript is of merit and in my opinion it could be considered for publication in the present form.

Best regards

Round 2

Reviewer 1 Report

I think the manuscript can be published in the present version.